## REVIEW ARTICLE

# DNA–protein crosslink proteases in genome stability

Annamaria Ruggiano [ID] [1] & Kristijan Ramadan [ID] [1✉]

Proteins covalently attached to DNA, also known as DNA–protein crosslinks (DPCs), are common and bulky DNA lesions that interfere with DNA replication, repair, transcription and recombination. Research in the past several years indicates that cells possess dedicated enzymes, known as DPC proteases, which digest the protein component of a DPC. Interestingly, DPC proteases also play a role in proteolysis beside DPC repair, such as in degrading excess histones during DNA replication or controlling DNA replication checkpoints. Here, we discuss the importance of DPC proteases in DNA replication, genome stability and their direct link to human diseases and cancer therapy.

Our genome is constantly exposed to various forms of DNA damage. DNA–protein crosslinks (DPCs) are common lesions and form when a protein—of any size and nature—becomes covalently bound to DNA after exposure to a physical or chemical crosslinker. They are bulky and impose physical obstacles to various DNA metabolic processes, such as DNA replication, repair, transcription and recombination (Fig. 1). Thus, the fidelity of DNA replication across DPCs and the accuracy of DPC repair pathways are pivotal in avoiding genomic instability, which can lead to ageing-associated diseases such as cancer. Whereas the consequences of unrepaired DPCs are well appreciated, the components, regulation and dynamics of the DPC repair pathway(s) are far from being understood.

DPCs are essentially removed by either canonical nucleases or dedicated proteases that degrade the protein component of the DPC (DPC proteolysis)[1,2]. DPC proteolysis has gained recognition with the discovery of the metalloproteases Wss1 (weak suppressor of Smt3-1) in yeast and SPRTN (SprT-like N-terminal domain, also known as Spartan or DVC-1) in higher eukaryotes[3–6]. Both proteases possess an intrinsic metalloprotease active center in specialized but phylogenetically distinguished domains (Fig. 2): the WLM (Wss1p-like metalloproteases) domain in Wss1 and SprT domain in SPRTN[1]. Wss1 and SPRTN depend on DNA binding for their proteolytic activity but have no defined sequence specificity. This feature is advantageous considering the heterogeneous nature of crosslinked proteins. However, it also exposes chromatin-associated and potentially functional proteins to the risk of undesired proteolysis. Thus, DPC proteolysis must be tightly regulated to minimize such risks. A comprehensive understanding of these regulatory modes is still lacking[7]. Remarkable progress in the field of

[1] Medical Research Council (MRC) Oxford Institute for Radiation Oncology, Department of Oncology, University of Oxford, Roosevelt Drive, OX3 7DQ Oxford, UK. ✉email: kristijan.ramadan@oncology.ox.ac.uk

**Fig. 1 Proteolysis protects cells from DPC toxicity during various DNA metabolic processes. a** DPCs on the leading or lagging strand can pose impediments for helicase and/or polymerase progression. **b** Proteolysis removes the bulk of this obstacle (DPC), and reduces the crosslinked protein to a peptide (remnant) that can be bypassed by TLS polymerase, thus resuming DNA replication. **c** DPCs can block transcription, but the mechanisms of transcription-dependent DPC proteolysis have not been explored.

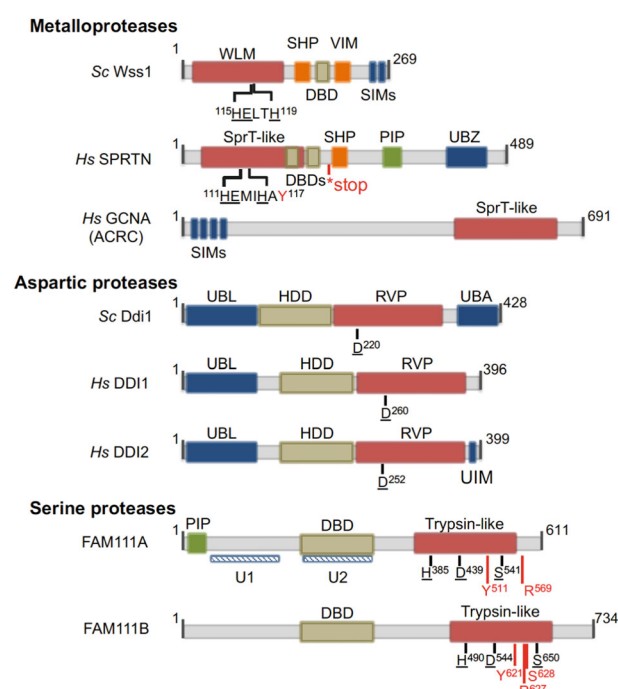

**Fig. 2 The structural features of the known DPC proteases.** Domains/motifs for interactions are color-coded. Red: protease domains. Orange: Cdc48/p97 interaction. Green: PCNA interaction. Ochre: DNA binding. Blue: SUMO/ubiquitin/proteasome binding. The catalytic residues in the protease domains are underlined. The sites of disease-associated mutations in SPRTN (RJALS)[60], FAM111A (KCS2)[118,119], and FAM111B (POIKTMP)[47] are marked in red. The asterisk in SPRTN indicates the site of nonsense mutations in RJALS patients that generate the protein SPRTN-ΔC[60]. RVP, retroviral protease. U1 and U2, potential UBLs in FAM111A[61].

DPC repair has also been made in recent studies on emerging DPC proteases with functional overlap to Wss1 and SPRTN. These proteolytic activities are primarily employed in DPC repair.

Besides DPCs, other types of substrates have been described for these proteases, for example tightly bound, albeit not crosslinked, chromatin proteins with the potential to disrupt the fidelity of DNA replication, like trapped poly (ADP-ribose) Polymerase-1 (PARP-1). It is therefore becoming increasingly clear that DPC proteases guard multiple aspects of genome maintenance.

In this review, we briefly discuss DPC formation, and expand on DPC proteases and the first and essential step of DPC repair: proteolysis of the protein component. We revise how DPC proteases contribute to genome maintenance, especially during DNA replication, and why defects in their activities lead to human disease.

## DNA–protein crosslinks

DPCs originate when proteins become crosslinked to DNA after exposure to physical or chemical agents, such as UV light or aldehydes, respectively (non-enzymatic DPCs), or as a result of faulty enzymatic reactions (enzymatic DPCs)[8]. Enzymatic DPCs are well exemplified by Topoisomerase-1 and Topoisomerase-2 cleavage complexes (Topo-1ccs, Topo-2ccs). During the physiological reaction of Topoisomerase on DNA, a transient, covalent intermediate (i.e., cleavage complex) forms between the catalytic tyrosine residue and the DNA phosphate group (phosphotyrosyl linkage). Stabilization of the cleavage complex (and formation of a DPC) can happen spontaneously if DNA is damaged, but is enhanced in the presence of poisons, e.g., camptothecin (CPT) or etoposide, for Topo-1 or Topo-2, respectively[9,10]. Notably, Topo-1/2 poisons are widely exploited in cancer chemotherapy[2,11]. Enzymatic DPCs also include crosslinks of DNMT1 (DNA methyltransferase 1) to the DNA methylation inhibitor 5-aza-2'-deoxycytidine (5azadC) incorporated into DNA[12,13], and of HMCES (5-Hydroxymethylcytosine binding, ES-cell-specific) to abasic sites in single-stranded DNA[14].

In the case of non-enzymatic DPCs, virtually any protein—of variable size, structure and nature—in the vicinity of DNA can be crosslinked. One of the most potent crosslinkers, formaldehyde (FA), is heavily present in the environment and produced endogenously via processes like lipid peroxidation, and DNA, RNA or histone demethylation[15–19]. Hence, FA release can occur in the surroundings of DNA, implying that DPCs form continuously and cells must constantly overcome DPC-induced toxicity.

Defective DPC repair leads to sensitivity to crosslinking agents, faulty DNA replication and cell cycle abnormalities, which pave the way for chromosomal instability and carcinogenesis in humans and mice[20]. Hence, multiple pathways work to ensure a proper response to these insults. Nuclease-dependent mechanisms, like nucleotide excision repair (NER) and homologous recombination, operate in both bacteria and eukaryotic cells by excising the DNA flanking the DPC[21–25]. However, NER seems to have a fairly limited role in overall DPC repair, as it can only remove small DPCs (8–10 kDa in size in mammalian cells)[5,23,26]. Topo-1/2ccs can be excised by dedicated tyrosyl-DNA phosphodiesterases, TDP1 and TDP2, which cleave the phosphotyrosyl linkage[27]; this is normally shielded and becomes accessible to TDPs after partial proteolysis or structural changes of Topoisomerases[28–32].

A much more pliable, less discriminate way to process DPCs is through proteolysis of the protein component operated by specialized DPC proteases, namely DPC proteolysis repair.

## DPC proteases

**Wss1 and SPRTN: the first members of a class of repair enzymes discovered**. DPC proteolysis repair was discovered in yeast with the metalloprotease Wss1[3]. A concomitant study in *Xenopus* egg extract hinted at a similar pathway in metazoans[33]. The dedicated enzyme in metazoans was later found to be SPRTN[4–6]. However, phylogenetic analysis has revealed that Wss1 and SPRTN are not orthologs but functional homologs[1,11,34]. The similarity between their sequences is limited within the protease domains—WLM and SprT—and around the active center typical of metalloproteases (HExxH), and they show 24% identity[5] (Fig. 2).

Wss1 and SPRTN confer resistance to FA, showing a general role in DPC repair[3–6]. This is also supported by in vitro cleavage studies, which demonstrate that Wss1 and SPRTN can process DNA-binding proteins of variable size and structure, exemplified by the smaller histone H3 and the larger Topo-2[3–6]. Although Wss1 is capable of processing purified Topo-1, *WSS1* deletion alone does not sensitize yeast cells to CPT, in striking contrast with the effects of SPRTN depletion in mammalian cells[3,5]. This is due to the existence of a yeast protease with a similar function and will be discussed below. A good explanation for this promiscuity came from structural studies of the yeast WLM domain[35]. As the protease domain lacks a substrate-binding pocket, it can accommodate a vast range of structures. DPC proteolysis leaves a remnant peptide, still of unidentified size, attached to DNA. Although less problematic for DNA replication and transcription, this remnant must be repaired via other activities (e.g., nucleases, tyrosyl-DNA phosphodiesterases) in coordination with or post-proteolysis. How this is achieved is not well understood.

Active DPC proteases are potentially dangerous and cells adopt strategies to restrain their activities[7]. First, Wss1 and SPRTN bind to and are activated by single-stranded (ss) and double-stranded (ds) DNA[3–6,36–38]; this feature protects nuclear soluble proteins from unwanted proteolysis. The structure of the SprT domain from human SPRTN suggests that the DNA dependence relies on the active site being shielded in the absence of DNA; DNA

**Table 1 Essentiality of genes encoding for DPC proteases.**

|  | CRISPR | RNAi | Embryonic lethality |
|---|---|---|---|
| *SPRTN* | 690/757 | 0/547 | Zebrafish, mouse[60,115] |
| *FAM111A* | 0/769 | ND | ND |
| *FAM111B* | 1/769 | 0/546 | ND |
| *ACRC/GCNA* | 1/721 | 1/285 | Fly, Worm, zebrafish[40] |
| *DDI1* | 0/769 | 6/710 | ND |
| *DDI2* | 128/769 | 17/710 | ND |

The numbers of dependent cell lines over the total tested are reported for each of the proteases, according to www.depmap.org. "CRISPR" and "RNAi" refer to data obtained after knockout or silencing, respectively. "Embryonic lethality" lists the publications that reported embryonic lethality after knockout or downregulation of the relative gene. *ND* not determined.

binding exposes the narrow groove with catalytic residues for peptide cleavage[37]. Second, Wss1 and SPRTN are capable of in *trans* self-cleavage in the presence of DNA; this property is a regulatory mechanism to shut down the protease activity when at the chromatin[4]. Third, the activity of DPC proteases is further sharpened by post-translational modifications (phosphorylation, ubiquitylation, SUMOylation, acetylation) of the DPC proteases and/or their substrates, and interaction with partner proteins, for instance the sliding clamp PCNA (proliferating cell nuclear antigen), the ATPase p97 or the Topo-1 binding protein TEX264. The latter point will be expanded in a separate paragraph.

**Emerging DPC proteases in genome stability**. Recently, proteases other than SPRTN have emerged as potential DPC repair enzymes. These are ACRC/GCNA, FAM111A and B, DDI1 and DDI2 in human, and their respective orthologs. Among the human enzymes, *SPRTN* is the only essential gene in variety of cancer cell lines (Table 1; www.depmap.org).

These emerging DPC proteases have been linked to processing of DPCs and tightly bound proteins (Figs. 2 and 3). Besides the protease domain, where conserved catalytic residues map (Fig. 2), they share remarkable similarities in overall domain organization, which underlies potentially similar patterns of regulation. Nonetheless, some proteases might be more specific for a certain cell cycle phase, a developmental stage, or a class of substrates.

*Germ cell nuclear antigen; also known as acid repeat-containing protein (ACRC)].* Germ cell nuclear antigen (GCNA) contains a SprT metalloprotease domain conserved across eukarya[11,39]. In metazoans, it is expressed in germ cells, where it was recently shown to target Topo-2ccs[40,41]. GCNA ectopically expressed in mammalian culture cells has also been involved in the resolution of 5-azadC-induced DNMT1 crosslinks[42] (Fig. 3). *GCNA* is not essential in mammalian cell lines (Table 1), but its knock-out (KO) increases the chances of embryonic lethality in flies, worms and zebrafish[40].

*DNA damage inducible 1 (DDI1) and DDI2 proteins.* DDI1 and DDI2 proteins are aspartic proteases that interact with ubiquitin and the proteasome via a ubiquitin-like domain (UBL)[43,44] (Fig. 2). They promote proteasome-dependent replication fork restart after replicative stress through degradation of replication termination factor 2 (RTF2)[44]. A direct involvement in DPC repair has not yet been explored, however the *S.cerevisiae* homolog Ddi1 was recently shown to aid Wss1 in resolution of CPT-induced and FA-induced DPCs[45,46].

*Family with sequence similarity 111 member A (FAM111A).* FAM111A and its homolog FAM111B are serine proteases.

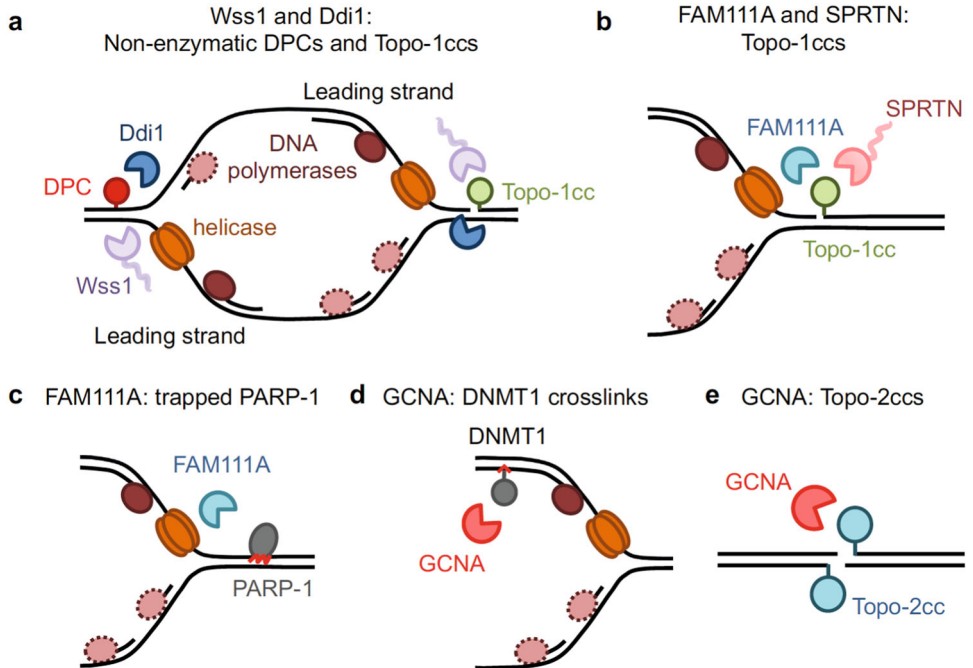

**Fig. 3 Overlapping and non-overlapping functions of DPC proteases. a** Wss1 and Ddi1 are both required for the repair of non-enzymatic DPCs and Topo-1ccs during replication. **b** FAM111A and SPRTN have both been implicated in Topo-1cc resolution, based on CPT resistance. **c** Sensitivity to PARP-1 inhibitors and DNA combing assays suggest that FAM111A, but not SPRTN, is involved in the resolution of trapped PARP-1. **d** Human GCNA resolves DNMT1 crosslinks at 5-azadC incorporation sites (red arrowhead), behind the replication fork. **e** GCNA resolves Topo-2ccs in flies, worms and zebrafish.

Not much is known about FAM111B, except for its association with the etiology of a form of poikiloderma with pulmonary fibrosis (POIKTMP)[47]. FAM111A was previously described as a PCNA interactor and for its role in restricting viral replication in host cells[48–50]. Only recently has it been implicated in the repair of Topo-1ccs and trapped PARP-1[51] (Figs. 3 and 4).

Evidence that GCNA, DDIs and FAM111A cleave DPC substrates is currently lacking. However, they do possess catalytic activity. FAM111A self-cleavage has been shown in vitro and in cells[51,52]; *Sc*Ddi1 and *Hs*DDI2 process substrates modified with long ubiquitin chains in vitro[53,54]. Moreover, complementation studies show that mutations in the putative catalytic residues of GCNA[40], *Sc*Ddi1[45], and FAM111A[51,52] generate non-functional proteins. For *Sc*Ddi1, the in vitro activity has been tested in the presence of dsDNA, an activator of Wss1 and SPRTN, without success[45]. Although DNA-dependent activity has been established as a regulatory mechanism for Wss1 and SPRTN, different proteases might respond to different modes of regulation.

A common feature among DPC proteases is DNA binding, a confirmation of their chromatin-related function. Minimal DNA binding domains (DBDs, Fig. 2) have been mapped by testing truncated versions in either DNA binding assays (e.g., EMSA)—as for Wss1, SPRTN, and FAM111A[3–5,37,51,55]—or complementation studies in cells—as for Ddi1's HDD (helical domain of Ddi1)[45,46]. Like SPRTN and Wss1, FAM111A can bind ssDNA, and the integrity of this domain is necessary for in vitro self-cleavage, as well as its function in cells[51]. A possible interpretation for the ssDNA dependent-proteolysis will be elaborated later.

**Overlapping functions of DPC proteases.** The existence of different DPC proteases indicates that cells have invested in multiple enzymes to counteract DPC-dependent toxicity (Fig. 3). This comes as little surprise considering the heterogeneous nature of crosslinked proteins. Although SPRTN does not have strict

sequence specificity, it preferentially cleaves DNA-binding proteins in poorly structured regions rich in lysine, arginine and serine residues[5]. The existence of alternative proteases would ensure that crosslinked proteins lacking these features are efficiently processed to prevent genomic instability. Here, we highlight the functional overlap between Wss1, SPRTN and the other proteases.

In metazoans, GCNA is predominantly expressed in germ cells and early embryos[39]. Therefore, most of the studies have been conducted in animal models rather than human cells. In *Drosophila* and *C.elegans*, *GCNA* mutation exposes germ cells and embryos to replication stress—formation of RPA/γH2Ax foci and hydroxyurea (HU) sensitivity—and genomic instability, e.g., chromosome segregation defects and micronuclei, which overall limit reproductive success[40,41]. *C.elegans* fertility defects are exacerbated by concomitant mutations in the *SPRTN* homolog *dvc-1*[40,41]. *Drosophila* embryos mutated in both *Gcna* and *mh* (maternal haploid, mh, is the *Drosophila* SPRTN homolog) do not complete embryogenesis[40]. Setting a role in DPC repair along with DVC-1/mh, *GCNA* deletion increases total DPCs in germ cells and early embryos of flies, worms and zebrafish[40], with Topo-2cc being among the most abundant[40,41]. *Dvc-1* and *gcna-1* mutant worm embryos are equally sensitive to formaldehyde[42]. Overall, this indicates an overlap between GCNA and DVC-1 at the organismal level. Germ cells and embryos are particularly vulnerable to the genomic instability resulting from DPC accumulation because mistakes would be inherited. Also, changes in gene expression and histone demethylation during embryogenesis[56] might especially expose germ cells to FA release[17] and DPCs formation, which explains the need for having multiple DPC proteases.

*S.cerevisiae* Ddi1 and human FAM111A are required for tolerance to Topo-1ccs, a common target of yeast Wss1 and human SPRTN. Unlike *wss1* mutation alone, the deletion of both *WSS1* and *DDI1* sensitizes yeast cells to CPT and identifies Ddi1

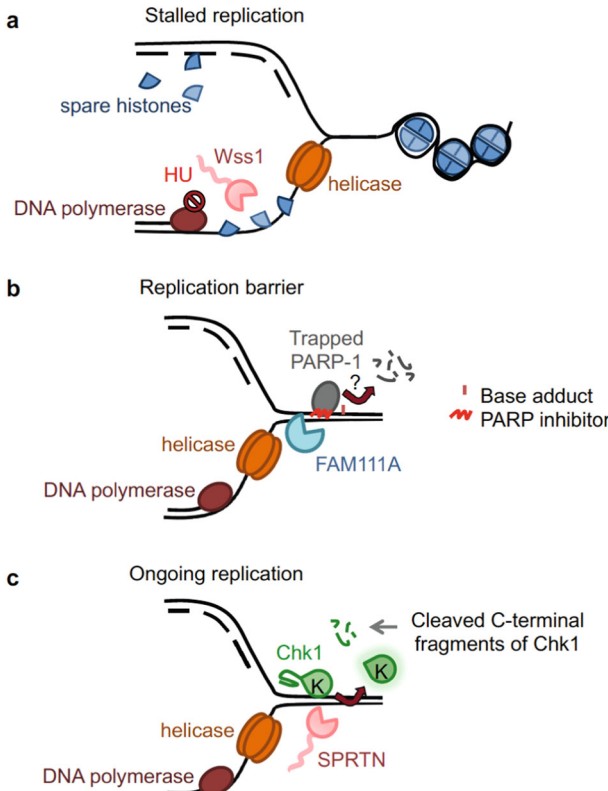

**Fig. 4 Proteolysis of non-DPC substrates during DNA replication. a** Wss1 cleaves excess histones binding to ssDNA after fork stalling. ssDNA is formed after HU. **b** PARP-1 is recruited to DNA to repair different types of damage, including base adducts generated by alkylating agents. In the presence of PARP inhibitors, self-PARylation and chromatin dissociation are blocked. Trapped PARP-1 represents a barrier for replication. FAM111A confers resistance to PARP inhibitors, but whether it acts proteolytically on PARP-1 awaits formal verification. **c** During DNA replication SPRTN cleaves the C-terminus of Chk1 and releases the active kinase domain (K) from chromatin.

as the elusive, redundant protease for Topo-1cc repair[3,45]. Genetic data additionally indicate that Ddi1 gives resistance to the DPC-inducing agent FA, along with Wss1[45,46].

SPRTN and FAM111A also have overlapping functions. Both proteins have been found at nascent DNA[5,50,57] and both prevent replication fork stalling in the presence of crosslinking agents, specifically, FA for SPRTN[5,58] and CPT for both FAM111A and SPRTN[51]. FAM111A depletion does not appear to reduce fork speed in unchallenged conditions, in contrast to SPRTN mutation[5,51,59–61]. Thus, FAM111A might come into play when the DPC overload exceeds the capacity of SPRTN, which is expressed at low levels. Such interplay is purely speculative at present. Interestingly, *FAM111A* KO but not *SPRTN* hypomorphic cells are sensitive to PARP-1 inhibitors, which were shown to trap PARP to DNA, suggesting that some DPC proteases might have preference for certain substrates[51].

**Regulation of DPC proteases by post-translational modifications**. Post-translational modifications (PTMs) have great regulatory potential, and most of the proteases discussed here can bind ubiquitin, via ubiquitin-associated domain (UBA) or ubiquitin-binding zinc finger (UBZ), or SUMO (Small ubiquitin-like modifier) via SUMO-interacting motif (SIM) (Fig. 2). Treatment of cells with DPC inducing agents, like FA and 5-azadC, induces signaling cascades, which culminate with the

modification of the crosslinked proteins by ubiquitin and SUMO[42,62,63]. We can predict that recruitment to or persistence on the damage site is modulated by PTMs. GCNA, for example, localizes to DNMT1 foci that form after 5-azadC treatment; this relocation depends on SUMOylation and GCNA's SIMs[42] (Fig. 2). SPRTN forms nuclear foci after exposure to FA and fails to do so upon pharmacological inhibition of the ubiquitylation pathway[42,62]. Whether this relocation depends on its UBZ remains to be demonstrated. This initial evidence, substantiated by the predominance of SUMO-interacting and ubiquitin-interacting domains, proves that DPC proteases have the potential to be regulated via interaction with post-translationally modified proteins.

DPC proteases can themselves be modified for regulatory purposes. SPRTN exists in a mono-ubiquitylated form, but exposure to FA leads to its de-ubiquitylation—two different studies have identified the de-ubiquitylating enzymes VCPIP and USP11—and acetylation to allow recruitment to chromatin[4,64,65].

Wss1 and SPRTN interact with the ATPase Cdc48/p97 also known as valosin-containing protein (VCP) in mammals, via SHP (suppressor of high copy PP1)-box and VCP-interacting motif (VIM)[3,36,66–68] (Fig. 2). Cdc48/p97 is a ubiquitin-dependent and SUMO-dependent chaperone acting with specific cofactors to unfold substrates for proteasomal degradation or disassembly from various macromolecules[69,70]. The unfolding activity of p97 directly aids repair of Topo-1ccs, to which p97 is recruited via its novel cofactor TEX264 (Testis Expressed 264) before SPRTN-mediated proteolysis[68]. Whether and how unfoldases are a constitutive requirement of DPC proteolysis repair remains to be established.

**DPC proteases and DNA replication**. DPCs can stall the CMG (Cdc45-Mcm2-7-Gins) replicative DNA helicase complex and/or DNA polymerase during DNA replication, depending on their size and location (leading or lagging strand)[33,71] (Fig. 1). Actively replicating cells are more susceptible to FA and CPT than non-replicating cells[5,72,73]. If left unrepaired, these replication obstacles can cause prolonged fork stalling that leads to fork collapse, formation of double-strand breaks (DSBs) and genomic instability[74].

In yeast, Wss1 has been proposed to allow completion of DNA replication after FA exposure[3], and Ddi1 is recruited to the damage site for removal of a model DPC in synchronized, S phase yeast cells[45]. However, repair of DPCs in a DNA replication-dependent manner has been more directly shown for SPRTN, both in human cells[5] and in *Xenopus* egg extract[75,76].

In line with S phase repair, SPRTN levels are higher in S/G2 phase—due to APC (Anaphase promoting complex)-Cdh1-dependent degradation in G1 phase[66]—when it localizes directly at the replication fork[5,57]. SPRTN interacts with PCNA via a PIP (PCNA-interacting protein)-box (Fig. 2), which is dispensable for SPRTN recruitment to the replication fork. A truncation variant lacking the C-terminus (and therefore the PIP-box) still localizes at nascent DNA in iPOND (isolation of proteins on nascent DNA) experiments[57] and rescues DNA fiber length[60]. Thus, it is unclear how SPRTN interacts with the replication machinery for DPC repair.

In vitro, substrate processing by SPRTN is most effectively stimulated by ssDNA—with the potential to form hairpins—or dsDNA structures with nicks or gaps close to the crosslinked protein[4,37,38]. This is consistent with results obtained in *Xenopus* egg extracts using a DPC plasmid, where SPRTN cleaves a model DPC without ongoing replication when there is a gap in the complementary DNA filament[75]. During DNA replication in cells, uncoupling between helicase and stalled DNA polymerase

exposes ssDNA and generates a dsDNA/ssDNA 'hybrid' that is compatible with the in vitro model. This specific DNA context presumably protects non-crosslinked chromatin-associated proteins from SPRTN activity[38].

GCNA is also active in S phase. Its role during DNA synthesis is supported by the replication stress (formation of RPA/γH2Ax foci) in mitotically active germ cells of GCNA mutant flies and worms[40], and by the fact that the Mcm2-7 components of the CMG helicase complex figure among the DPCs in GCNA mutant fly embryos[40]. Another study links GCNA to the resolution of DNMT1 crosslinks, although this was shown with ectopically expressed GCNA in human somatic cells[42]. DNMT1 crosslinks form behind the replication fork, when DNMT1 is recruited to newly synthesized DNA to restore the DNA methylation pattern, where it can get crosslinked in the presence of 5-azadC. It remains to be determined whether effective DNMT1 crosslink repair needs GCNA catalytic activity.

FAM111A levels increase in late S phase and remain high during G2/M[48]. FAM111A interacts with PCNA via a PIP-box (Fig. 2)[50,52]. It is unclear to what extent FAM111A depletion impairs DNA replication. EdU incorporation studies have produced conflicting results[50,52,61]. DNA combing assays in the absence of exogenous damage have either shown no reduction in the length of nascent DNA in FAM111A-depleted cells or a slight increase in tract length to compensate for the decrease in origin firing[51,61]. Instead, FAM111A KO reduces replication speed upon treatment with CPT and PARP inhibitors and increases cellular sensitivity to these poisons[51]. Consistently, Topo-1cc foci accumulate in KO cells; an intact PIP-box and an intact protease domain are required to rescue these phenotypes[51]. Although in vitro proteolysis of Topo-1 has not been shown so far, FAM111A may be required for the removal of replication barriers. In this respect, the extent of overlap with SPRTN remains to be determined.

Strikingly, FAM111A overexpression causes EdU incorporation and cell cycle defects, and consequently increases the rate of DNA damage and apoptosis; this is probably due to unspecific cleavage of replication fork proteins, e.g., PCNA and RFC (Replication factor C)[50,52,61,77]. However, overexpression of the PIP-box mutated variant recapitulates these phenotypes to some extent, arguing that binding to PCNA might not be the cause of the replication problems[50,52,61]. These replication defects are exacerbated by overexpression of FAM111A variants carrying disease-causing mutations (Kenny–Caffey syndrome type 2, see below and Fig. 2)[52,61,77]. Overexpression of FAM111B mutant variants, causative of POIKTMP (Fig. 2), largely mimics the phenotype of FAM111A overexpression[52]. In both cases, concomitant mutations in the protease domains abolish these defects, showing that the observed phenotypes are the direct cause of FAM111A/B-dependent proteolysis[52]. These data suggest that deregulated FAM111A/B enzymatic activity is detrimental to the cells. Thus, tight regulation of the level and activity of these proteases is critical for the successful completion of DNA replication.

*Downstream pathway and translesion DNA synthesis.* Translesion DNA synthesis (TLS) is a damage tolerance mechanism where the conventional DNA polymerase is exchanged for an error-prone TLS polymerase that bypasses the lesion. This switch depends on PCNA and its ubiquitylation by the E3 ubiquitin ligase Rad18[78]. SPRTN was linked to TLS regulation after UV damage before its role in DPC repair was established[66,67,79–82]. SPRTN interacts with both PCNA and ubiquitin, but how SPRTN regulates TLS is not entirely clear. Some studies suggest it is a positive regulator of TLS[79,81,82], while others propose that SPRTN prevents UV-induced, TLS-dependent mutagenesis[66,67,80].

Although TLS regulation by SPRTN was initially linked to UV lesion repair, it is plausible that bypass of the remnant peptide after bulk DPC proteolysis also requires TLS (Fig. 1). Some lines of evidence support this: (i) UV light can crosslink proteins to DNA[83–85]; (ii) epistasis analysis shows that Rad18 and SPRTN work together[58]; (iii) in Xenopus egg extracts, TLS polymerases are required to complete replication of a plasmid carrying a model DPC after digestion by SPRTN[75]; (iv) in yeast, WSS1 deletion lowers mutation rates after FA[3]. Therefore, TLS might act immediately after DPC proteolysis to avoid fork stalling, since mutations are still more desirable than replication fork collapse.

There is one case where the accumulation of mutations can be beneficial. This is the hypermutation of immunoglobulin loci. SPRTN has been shown to favor mutations (DNA polymerase eta and zeta-dependent) at abasic sites and template switch-mediated gene conversion during immunoglobulin gene diversification in chicken B cells[86]. Notably, abasic sites are a common source of DPC formation, and the hypermutation phenotype is consistent with TLS acting downstream of DPC proteolysis. In agreement, lower mutation rates at the immunoglobulin locus are observed in SPRTN-depleted cells[86]. This suggests that SPRTN plays a role in generating the diversity of immunoglobulin repertoire and in the DNA damage tolerance pathway when DNA replication forks get stalled.

**DPC repair outside S phase**. Despite the numerous bodies of evidence linking SPRTN to replication-coupled DPC proteolysis, a replication-independent function of SPRTN has not been ruled out[4,5]. SPRTN mutations or insufficiency manifest into hepatocellular carcinoma[57,60]. The liver might be more exposed than other organs to DPCs because inhaled and ingested compounds can be catabolized into formaldehyde. However, the great majority of hepatocytes are quiescent[87], therefore it is plausible that SPRTN is operating outside of S phase, and that malfunctioning would lead to cancer. It is also possible that liver damage observed in SPRTN-mutated patients or SPRTN hypomorphic mice initiates during embryogenesis' S phase but this defect emerges later.

Starvation-arrested worm larvae, where replication is not happening (cells are arrested in G1/S), are sensitive to formaldehyde[4], indicating that processes other than DNA synthesis are affected. Another example of SPRTN activity outside S phase comes from Drosophila. The expression of a mutated mh (the SPRTN homolog) from the maternal genome in the zygote leads to loss of the paternal DNA during the first mitotic division and results in the lethality of haploid embryos—hence the name mh, "maternal haploid"[88,89]. This loss is caused by the lack of condensation of the paternal DNA during mitosis, and it is inherently unlinked to DNA replication and S phase[88]. The paternal DNA remodeling still relies on mh catalytic activity, but the mechanism is unclear[88,89].

GCNA also works outside S phase. Its expression peaks in mitosis, where it localizes on chromosomes[41]. Topo-2ccs are prominent GCNA targets[40,41]. Topo-2ccs are abundant in mitosis because Topo-2 activity is required for separation of sister chromatids. Supporting a role for GCNA in Topo-2cc repair, GCNA and Topo-2 physically interact and colocalize, and mutant worms are sensitive to etoposide but not to camptothecin[41].

Overall, these data confirm that DPC protease activity is essential to maintain genome integrity during DNA replication, as well as outside of S phase.

**DPC proteases and transcription**. DPCs are also potential obstacles for transcription[20]. Although transcriptional stalling and mutagenesis can be threatening for genome stability[90],

the effects of DPCs on the progression of RNA polymerases are a largely neglected field.

Some evidence linking DPC proteases and transcription has emerged. Rpb1, the largest and catalytic subunit of the RNA polymerase II, is degraded via Ddi1 and Wss1 after exposure to HU or UV light[45]. Both proteases interact with Rpb1. However, this evidence provides insufficient proof that Ddi1 and Wss1 target Rpb1 as a crosslinked protein. Its degradation might be a consequence of transcription stalling due to roadblocks ahead of RNA polymerase II. In fact, under these circumstances Rpb1 degradation was described to rely on the ubiquitin-proteasome system, the ATPase Cdc48/p97 and SUMO[91,92]. At any rate, a stalled, non-crosslinked Rpb1 would be equally "eligible" for degradation by DPC proteases, since other "non-DPC" substrates exist and will be discussed later in this review.

A link between transcription and FAM111A in human cells also exists. FAM111A overexpression reduces Rpb1 levels at the chromatin and, consequently, EU incorporation[52]. FAM111A also interacts with Rpb1 upon overexpression[52]. However, it is unclear if these phenotypes are related to direct Rpb1 proteolysis.

**DPCs and proteasome**. The most studied proteolytic machinery in the cell is the 26S proteasome (henceforth proteasome), which degrades proteins that have been previously modified by ubiquitin and unstructured by unfoldases[93,94]. It is not surprising that proteasome activity has been tested towards DPCs[26]. The involvement of the proteasome implies that the DPC must be modified with ubiquitin. Early sources of evidence showed that proteasome inhibition delayed Topo-1/2 degradation following exposure to Topo-1 and −2 poisons, suggesting that the proteasome is required for Topo-cc repair[95–97]. However, these initial experiments failed to substantiate the formation of ubiquitylated Topo-ccs. Topoisomerase ubiquitylation was shown in whole-cell extracts – rather than on chromatin or Topo-ccs – and there was no increase in the topoisomerase ubiquitylation state after proteasomal inhibition[95,97]. Moreover, these studies used very high doses of CPT (μM range)[96,97], which crosslink 90% of Topo-1[98] and might cause an overload of Topo-1ccs beyond the repair capacity of the other, replication-dependent, proteases.

More recent studies have directly shown that both enzymatic and non-enzymatic DPCs are modified by ubiquitin[14,42,62,63,75]. However, whether this modification leads to proteasomal degradation seems controversial, as ubiquitylation might rather have a signaling role[62].

Crosslinked proteins can be concomitantly modified by ubiquitin and SUMO (SUMO-1 or SUMO-2/3)[42,62,63,75]. Modification via SUMO-2/3 prior to ubiquitylation hints at the involvement of SUMO-targeted ubiquitin ligases (STUbLs), which ubiquitylate SUMO-modified substrates for proteasomal degradation[42,63]. However, scenarios have been described where these two modifications are independent, although the function of DPC SUMOylation in these cases remains unclear[62,75].

*S. cerevisiae* Ddi1 is a proteasome adapter[99] (Fig. 2). Ddi1's UBL binds the proteasome subunit Rpn1[100,101], and, unlike conventional UBL domains, it also binds ubiquitin, along with UBA[43,102]. However, there is no evidence that Ddi1 functions in DPC repair with the aid of the proteasome. In fact, over-expression of mutated versions lacking UBL and UBA domains complements *DDI1* deletion in FA sensitivity experiments, while mutation of the putative catalytic domain does not[45]. A different scenario is very likely for the human homologs DDI1 and DDI2. DDI1/2 bind the proteasome via their UBLs[43,44] (Fig. 2). DDI1/2 recruit the proteasome at stalled replication forks to remove RTF2, whose persistence would prevent fork restart after HU and cause chromosome instability[44].

As for proteasome-dependent DPC repair during replication, experiments in *Xenopus* egg extracts have provided the most compelling evidence[75,76]. Instead, in mammalian cells, ubiquitylation of Topo-ccs via the STUbL RNF4 prior to proteasomal degradation is independent of replication[63]. A recent report claims that the enzyme HMCES crosslinks to abasic sites ahead of the replication fork. In this way, HMCES shields the abasic sites from replication by TLS polymerases and prevents mutagenesis[14]. HMCES crosslinks are ubiquitylated and processed by the proteasome, but the role of other proteases has not been tested[14]. Although physiological crosslinking must happen during DNA synthesis to prevent switching to TLS polymerases, it has not been established that HMCES-DPC repair by the proteasome happens during replication. Post-replicative repair is plausible, as (i) HCMES-DPC per se represents an impediment to DNA polymerase progression and (ii) proteolysis by the proteasome would likely leave a peptide that can halt the DNA polymerase. In either case, a tolerance mechanism other than TLS must be in place to bypass the DPC or the remnant peptide (e.g., template switching). Therefore, post-replicative proteasome-dependent proteolysis remains a formal possibility in this case.

In conclusion, replication-coupled proteolysis by a large complex like the proteasome remains a matter of future investigation in mammalian cells. For example, it will be informative to determine whether proteasome subunits appear at nascent chromatin after treatment with DPC-inducing agents (e.g., by iPOND), since assessing the proteins enriched at stalled forks has been useful for other kinds of replication stress[103].

**DPC proteases in processing of non-crosslinked, tightly bound substrates**. Mass spectrometry has been performed on DPCs isolated from cells depleted of SPRTN[5] and GCNA[40] to identify the most abundant crosslinked proteins. These screenings have proved that Topoisomerases, histones and Mcm subunits of the CMG helicase complex are the major DPCs in SPRTN-depleted cells and *GCNA* KO flies[5,40]. Nucleic acid-binding proteins are expected to come up in such screenings[5] since proteins acting in the vicinity of the DNA are most likely crosslinked. However, in vitro cleavage experiments are mainly performed on substrates that have the propensity for DNA binding but are not crosslinked to DNA[3–6,37]. Hence DNA association rather than crosslinking per se is a requisite for cleavage. Recent studies have proven that this is true in cells as well. The three studies below illustrate that proteases process non-covalently DNA-associated proteins to protect cells from DNA replication errors.

*Histones:* A recent study postulates that Wss1 degrades unassembled, yet non-covalently bound, histones during replication stress[104]. This Wss1 activity would protect cells from the unspecific binding of excess histones to the ssDNA accumulating after HU exposure, which can interfere with DNA metabolism[104] (Fig. 4A). The sensitivity of *wss1* mutant yeast cells to HU is greatly enhanced by concomitant deletion of *DDI1*, and a catalytically inactive Ddi1 does not rescue cell survival[45,46]. Thus, the protease Ddi1, like Wss1, might be required to cope with replication stress beyond DPCs. More interestingly, complementation studies in yeast suggest that hDDI1/2 retain the same function[46].

*PARP-1:* Tightly bound proteins can be as dangerous as DPCs for replication fork progression[105]. This scenario is well illustrated by PARP-1 trapping. PARP-1 is an enzyme participating in DNA replication and many DNA repair pathways[106,107]. PARP-1 catalyses the formation of poly-ADP ribose (PAR) chains—a reaction known as PARylation—that recruit other repair proteins; self-PARylation triggers an electrostatic dissociation from chromatin, allowing the repair reaction to proceed[106]. Pharmacological inhibition causes

PARP to become tightly bound—or trapped—on chromatin because of the defective self-PARylation. Chromatin persistence due to PARP inhibitors is more problematic than the catalytic inhibition per se[108]. In fact, trapped PARP-1 can cause replication fork collapse and DSBs and has the potential of killing *BRCA*-deficient cells, a strategy that is used for treatment of cancer patients with *BRCA* mutations[109]. FAM111A depletion sensitizes cells to niraparib and talazoparib[51], two PARP inhibitors with a strong trapping capacity[108,110]. Niraparib treatment reduces replication fork speed in FAM111A-depleted cells and causes replication stress. Rescue of the replication defect depends on FAM111A catalytic residues, PIP-box and DNA binding, implying that FAM111A uses its proteolytic activity to remove obstacles imposed by trapped PARP-1 to replication forks[51] (Fig. 4B). Whether other proteases participate in the removal of tightly bound proteins remains to be established.

*Chk1:* The degradation of a non-crosslinked substrate echoes DPC removal: in either case, proteolysis eliminates potential obstacles for DNA replication and metabolism, which threaten cell viability[51,104]. In striking contrast is the cleavage of the checkpoint kinase Chk1 by SPRTN during DNA synthesis[59]. During S phase, SPRTN cleaves Chk1 at the chromatin, in a loosely structured region at the C-terminus, and releases truncated Chk1 N-terminal fragments with stronger kinase activity than the full-length protein[59] (Fig. 4C). This way SPRTN ensures a basal and physiological Chk1 activation to support DNA replication progression in the absence of exogenous damage, i.e., in the absence of insults that cause ssDNA accumulation and robust ATR-dependent Chk1 activation cascade[59,111]. Overexpression of Chk1 N-terminal fragments restores normal DNA replication in SPRTN-depleted cells (i.e., replication fork speed and origin firing) and partially rescues developmental defects of *SPRTN*-deficient zebrafish embryos[59].

Besides uncovering an important role for Chk1 during unperturbed replication, this study also shows that the function of DPC proteases goes well beyond the proteolysis of replication fork barriers.

**DPC proteases and diseases**. Defective DNA replication, often referred to as DNA replication stress, is one of the major causes of cancer[112–114]. DNA replication stress can be triggered by obstacles ahead of the replication fork, such as those imposed by crosslinked or tightly bound proteins. Since the activity of DPC proteases is linked to DNA replication, strong association exists between defective DPC proteases and human diseases[40,57,60].

*Ruijs-Aalfs syndrome (RJALS).* Biallelic and monogenic mutations in *SPRTN* cause a rare autosomal recessive progeroid disease known as RJALS[60]. This was the first disease to be linked to defective DPC proteolysis repair. The loss-of-function mutations identified in patients so far are a missense mutation that generates the catalytically inactive SPRTN-Y117C variant, and non-sense mutations that generate a truncated SPRTN, SPRTN-ΔC (Fig. 2), which retains partial functionality due to intact protease activity but defective cellular localization[4,5,57,60]. RJALS patients show signs of premature aging—including cataracts, graying of the hair, lipodistrophy—and develop early-onset hepatocellular carcinoma (HCC)[60]. These phenotypes have been recapitulated in *SPRTN* hypomorphic mice[57,115], and definitively prove that defective SPRTN alone is responsible for the RJALS phenotype.

Aging and cancer are seemingly conflicting manifestations, but truly two outcomes of the same underlying cellular defects—the accumulation of DNA damage and genomic instability[116,117]. RJALS patient cells (patient-derived lymphoblastoid cell lines and fibroblasts) display DNA replication stress, G2/M leakage, increased number of DSBs, chromosomal aberrations and increased level of total DPCs. It is unclear why human patients and mice develop HCC while the rest of the body ages.

*Kenny–Caffey syndrome type 2 (KCS2) and gracile bone dysplasia (GCLEB).* FAM111A mutations cause KCS2, a rare autosomal dominant disease characterized by short stature, hypoparathyroidism, hypocalcemia and abnormal bone development. KCS2 patients do not show any predisposition to develop cancer[118,119]. Heterozygous mutations in *FAM111A* have also been found in patients with GCLEB, or osteocraniostenosis[119]. GCLEB is lethal in newborns because of severe skeletal abnormalities.

KCS2-causing and GCLEB-causing mutations alter FAM111A proteolytic activity. Substitution of R569 to Histidine (H) is a recurrent mutation in KCS2[118,119] and falls in the proximity of the catalytic residue S541 (Fig. 2). FAM111A-R569H and other disease-causing mutations enhance FAM111A self-cleavage in vitro, showing that they are gain-of-function mutations[51,52]. Expression of FAM111A-R569H or other mutated variants causes replication and transcription defects and apoptosis[52,61,77]. Thus, it seems plausible that the toxicity of the hyperactive protein arises from uncontrolled degradation of replication and transcription proteins (e.g., RFC, Rpb1)[52]; however, it is unknown how these mutations result in the observed phenotypes.

*Pediatric germ cell tumors (GCTs).* Mutations in *GCNA* associate with GCTs. Downregulation of *GCNA* expression—due to copy number loss and promoter hypermethylation—is found in 66% of GCTs[40] and correlates with a poor prognosis. These alterations in GCTs show that GCNA protease is critical for the genomic stability of germ cells.

## Conclusions and perspectives

The topic of DPC proteolysis repair has undergone an extraordinary advance in the past year. The list of specific repair enzymes has come to include novel DPC proteases working with Wss1 and SPRTN. In most cases, direct proteolysis of DPC substrates remains to be formally tested with in vitro assays; however, experiments conducted in cells, such as sensitivity to crosslinking agents, have established that an intact catalytic domain is always essential. We anticipate that more rigorous and systematic in vitro studies will elucidate the optimal requirements for DPC proteolysis.

The fidelity of DNA replication and genome stability are compromised by loss or misregulation of these proteases. This is not only due to defective DPC repair but also to defective removal of non-covalent replication barriers. The latter exemplifies other important functions of DPC proteases in the maintenance of genome integrity.

DPC toxicity opens an interesting perspective for cancer treatment as well. The most commonly used chemotherapy agents are Topoisomerase poisons that induce enzymatic DPCs known as Topo-1/2ccs[2,11]. Specific DPC protease inhibitors could synergize with DPC-inducing agents and be instrumental to chemotherapy. The recent finding that FAM111A confers resistance to PARP inhibitors[51] has potential implications in cancer treatment, not only in light of the development of a FAM111A inhibitor. In fact, FAM111A expression varies among cancer cell lines[51], thus the efficacy of PARP inhibitors might change. These types of information should help design better treatment strategies.

We expect that research will focus more imminently on the identification of substrates and the regulatory mechanisms, including interaction and coordination with other repair proteins for timely activation of the DPC protease activity. This basic knowledge of DPC proteases will pave the way for a better understanding of human diseases and cancer therapy.

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

## Acknowledgements

We deeply apologize to the researchers whose work was not cited. We thank a Medical Research Council (MRC) Programme grant to K.R. (MC_EX_MR/K022830/1). A.R. was supported by an EMBO LTF (ALTF 1109-2017). We would like to thank our DPhil student Gwendoline Hoslett for helping us with the text editing and her comments.

## Author contributions

A.R. wrote the draft and prepared the figures. K.R. initiated and supervised this work. A.R. and K.R prepared the final version of this manuscript.

## Competing interests

The authors declare no competing interests.
