## [Peer Review File · Communications Biology]

Reviewers' comments:

Reviewer #1 (Remarks to the Author):

This review focuses on DPC repair by proteases. This topic has been reviewed previously, but the perspective of these authors is valuable. I offer the following comments as suggestions for improvement:

I found the review difficult to read partly because of the organization. There is a lot of bouncing back and forth from topic to topic and the authors revisit the same point multiple times. For example, the links to diseases are stated in multiple locations even though there is an entire section with the title "Proteases and Disease". This is just one example where the titles of the different sections do not actually help the reader know what to expect (or not expect) to find in that section.

The issue of substrate specificity is confusing throughout the text. In some cases the authors suggest there is no specificity of these proteases and then in some cases suggest there is specificity. Clarification is needed to understand what the take home message is supposed to be. Are some of the proteases substrate specific? Do they have preferences? Is it like TLS polymerases where there is overlapping specificity?

Figure 2 could be improved to better illustrate of how proteolysis protects from DPC toxicity.

A figure illustrating the idea of redundant vs. non-redundant functions of the proteases could be useful.

Pg. 2 , abstract: Please change "cells have evolved" since statements like this about cells evolving things are not accurate. Likewise, the statement that Wss1 and SPRTN are "founders" is confusing. Is this meant to indicate some evolutionary statement or simply that they were identified first?

Pg. 3, intro: What is meant by "they are usually bulky"? Bulky in the sense of large? In comparison to most lesions? Wouldn't all DPCs fit this definition?

Page 3: "DPCs are essentially removed by nucleases" that "degrade the protein component" is incorrect.

Page 4: what is meant by "is an excellent resort towards"

Page 12: The statement "overall this indicates redundancy..." doesn't fit with the listed phenotypes showing that the mutants each causes sensitivity to formaldehyde. That indicates they are not redundant.

Pg. 17: I don't understand why lower mutation rates would be expected in SPRTN-depleted cells? I this because a pathway that doesn't involve proteolysis would be used? Some explanation is needed.

Reviewer #2 (Remarks to the Author):

In this review, the authors discuss recent progress in proteases involved in DPC repair. Author's group has published a number of key papers on SPRTN and contributed significantly to the current understanding of DPC proteases. Although Dr. Ramadan has authored several review articles on a similar topic in the last several years, this review covers new DPC proteases (Ddi1/2 and FAM111A/B) and therefore provide new insight into the roles of DPC proteases in the maintenance of genome stability.

Major points

1. Page 3, line 21: It might be useful to mention that there is a mechanism independent of proteases and nucleases. Removal of TOP2ccs by TDP2 assisted by ZATT does not fall into those two mechanisms.
2. Page 5, line 21: Because DPCs containing HMCES are discussed later in this review, it might be helpful to list HMCES DPCs here.
3. In Page 7, the authors discuss mechanisms that restrain SPRTN activities. SPRTN function is also restricted by chromatin accessibility through mono-ubiquitination. Given the recent paper (<https://doi.org/10.1016/j.molcel.2020.06.027>) and the preprint (<https://doi.org/10.1101/2020.06.30.180471>) describing deubiquitinases for SPRTN (VCPIP1 and USP11), it might be useful to discuss this regulatory mechanism here or as a separate section.
4. Page 9, line 8: Reference #47 does not seem to show DDI1/2's function in removal of non-covalent replication barriers as stated here. The referred paper suggested that RTF2 is a replisome component that needs to be degraded in response to replication stress, but did not claim that RTF2 is a replication barrier.
5. Page 13: The section of "DPC proteases and DNA replication" is well-written, but the role of the proteasome (<https://doi.org/10.1016/j.molcel.2018.11.024>) is not mentioned.
6. Page 14: line 22: Contrary to what is discussed here, Reference #58 seems to show CldU tract length to assess replication speed.
7. Page 24, line 7: "Analogous phenotypes are observed after exposure of human cells to FA or CPT, indicating that they are due to defective DPC repair." This might not be a valid argument. FA causes DNA crosslink damage and CPT might generate dsDNA ends, both of which do not involve DPCs but could cause phenotypes analogous to SPRTN depletion.

Minor points

1. Page 3, line 22: change "nucleases" to "proteases"
2. Page 7, line 20 – Page 8, line 13: This part seems to be out of place. The authors might consider combining with the following section of "Emerging DPC proteases."
3. Page 8, line 19: Protease names are indicated in bold (here ACRC/GCNA), but it does not seem to be following a unified rule. ACRC/GCNA shows up again in bold in Page 11, and other proteases are indicated in bold in Page 12 for the second time.
4. Page 23, line 2: Consider revising the sentence. (Rescue of) the replication defect depends on ...
5. Page 26, line 3: replace "than" with "that"

Reviewer #3 (Remarks to the Author):

In the manuscript, Ruggiano and Ramadan summarize latest advances in the field of DNA – protein crosslink (DPC) repair by proteases in relation to genome stability. The article begins with a general introduction on DPCs and follows by discussion on recently characterized DPC proteases and their known regulatory mechanisms. Thereafter, the review brings about recent important observations on the DPC-related proteases GCNA, SPRTN, and FAM11A in regulating DNA replication and genome

stability across species, as well as the roles of SPRTN and FAM111A against DPC during DNA replication. Beyond the DNA replication, the authors also elaborate on reported roles of DPC proteases on mitosis, transcription, and other known non cross-linked substrates (i.e. histones, PARP1, and CHEK1).

Overall, the review is well structured and provides concise update on the current knowledge on DPCs and DPC proteases.

However, in addition to further proofreading, the present review article may benefit from the following:

1. To include further description of DPC repair by NER on page 6, prior to the "Wss1 and SPRTN: the founding members of a class of repair enzyme"
2. To highlight the emerging post-translational regulatory aspects (i.e. SUMO and ubiquitin) of DPC repair (i.e. page 19 - 21). Authors may also consider to incorporate the discussion together with discussion DPC proteases structure and activation by DNA on page 6 & 7
3. To improve Figure 2 in illustrating different cellular aspects influenced by DPC is highly recommended
4. To check and remove duplicated reference from the review - e.g. ref 74 & 98 are duplicate

Point-by-point response to reviewers

We would like to thank the referees on their comments how to improve our manuscript entitled “**DNA-protein crosslink proteases in genome stability**” (COMMSBIO-20-1800). We have prepared the revised manuscript according to the reviewers’ suggestions/criticisms. We believe that have addressed all their comments and prepared the significantly improved version of our manuscript.

Reviewer #1 (Remarks to the Author):

This review focuses on DPC repair by proteases. This topic has been reviewed previously, but the perspective of these authors is valuable. I offer the following comments as suggestions for improvement:

I found the review difficult to read partly because of the organization. There is a lot of bouncing back and forth from topic to topic and the authors revisit the same point multiple times. For example, the links to diseases are stated in multiple locations even though there is an entire section with the title “Proteases and Disease”. This is just one example where the titles of the different sections do not actually help the reader know what to expect (or not expect) to find in that section.

Answer: We appreciate this comment and we have avoided this redundancy among paragraphs and reduced overall work count while doing so.

The issue of substrate specificity is confusing throughout the text. In some cases the authors suggest there is no specificity of these proteases and then in some cases suggest there is specificity. Clarification is needed to understand what the take home message is supposed to be. Are some of the proteases substrate specific? Do they have preferences? Is it like TLS polymerases where there is overlapping specificity?

Answer: We understand that confusion might have arisen. The message we want to deliver is that the two most studied DPC proteases, Wss1 and SPRTN, do not have sequence/substrate specificity. For other proteases (e.g. FAM111A), only specific substrates have been identified so far. This does not mean that they have substrate specificity. To convey a clearer message we have changes the wording as appropriate. For example, page 13, paragraph “Overlapping functions of proteases”: FAM111A, but not SPRTN, repairs trapped PARP-1. We replaced the expression “substrate specificity” with “preference for some substrates”.

Figure 2 could be improved to better illustrate of how proteolysis protects from DPC toxicity.

Answer: Figure 2 has been modified and it is Figure 1 now. New Figure 1 shows that DPCs can halt different processes (this improvement was also advised by Reviewer 3) and highlights the effect of proteolysis on DPC replication progression.

A figure illustrating the idea of redundant vs. non-redundant functions of the proteases could be useful.

Answer: We have added Figure 3 to show redundant and non-redundant functions.

Pg. 2, abstract: Please change “cells have evolved” since statements like this about cells

Point-by-point response to reviewers

evolving things are not accurate. Likewise, the statement that Wss1 and SPRTN are “founders” is confusing. Is this meant to indicate some evolutionary statement or simply that they were identified first?

Answer: “Cells have evolved” has been changed to “cells possess”. “Founding members” means they were the first to be identified. We have changed to “firstly discovered” to avoid confusion.

Pg. 3, intro: What is meant by “they are usually bulky”? Bulky in the sense of large? In comparison to most lesions? Wouldn’t all DPCs fit this definition?

Answer: Most DPCs would agree to the definition of being “bulky”, *i.e.* large. We have removed “usually”.

Page 3: “DPCs are essentially removed by nucleases” that “degrade the protein component” is incorrect.

Answer: This has been corrected.

Page 4: what is meant by “is an excellent resort towards”

Answer: We have changed this expression to “is advantageous” for clarity.

Page 12: The statement “overall this indicates redundancy...” doesn’t fit with the listed phenotypes showing that the mutants each causes sensitivity to formaldehyde. That indicates they are not redundant.

Answer: We agree with this comment. We intended “overlap” instead of “redundancy”. It has been corrected.

Pg. 17: I don’t understand why lower mutation rates would be expected in SPRTN-depleted cells? Is this because a pathway that doesn’t involve proteolysis would be used? Some explanation is needed.

Answer: we better clarify this in the text now (page 15). SPRTN works together with DNA pols eta and zeta in TLS and mutagenic template switch gene conversion at abasic sites and sites where DNA replication fork stalls. However, it is not known if SPRTN proteolysis is important for these functions or not.

Reviewer #2 (Remarks to the Author):

In this review, the authors discuss recent progress in proteases involved in DPC repair. Author’s group has published a number of key papers on SPRTN and contributed significantly to the current understanding of DPC proteases. Although Dr. Ramadan has authored several review articles on a similar topic in the last several years, this review covers new DPC proteases (Ddi1/2 and FAM111A/B) and therefore provide new insight into the roles of DPC proteases in the maintenance of genome stability.

Major points

1. Page 3, line 21: It might be useful to mention that there is a mechanism independent of

Point-by-point response to reviewers

proteases and nucleases. Removal of TOP2ccs by TDP2 assisted by ZATT does not fall into those two mechanisms.

Answer: We have taken the reviewer's suggestion and added a sentence on TDP1 and 2 at page 6 (adding at page 3 as suggested would have been premature, since we had not mentioned Topoisomerase crosslinks by that point). We added reference Pommier *et al.*, Tyrosyl-DNA phosphodiesterases (TDP1 and TDP2), DNA repair 2014. We also cited in this context:

Yang et al., PNAS 1996
Debethune et al., NAR 2002
Interthal et al., JBC 2005
Schellenberg et al., Science 2017
Gao et al., JBC 2014

2. Page 5, line 21: Because DPCs containing HMCES are discussed later in this review, it might be helpful to list HMCES DPCs here.

Answer: We agree. We have changed the text to accommodate a description on HMCES crosslinks.

3. In Page 7, the authors discuss mechanisms that restrain SPRTN activities. SPRTN function is also restricted by chromatin accessibility through mono-ubiquitination. Given the recent paper (<https://doi.org/10.1016/j.molcel.2020.06.027>) and the preprint (<https://doi.org/10.1101/2020.06.30.180471>) describing deubiquitinases for SPRTN (VCP1 and USP11), it might be useful to discuss this regulatory mechanism here or as a separate section.

Answer: We agree. We have included and discussed these two studies in our manuscript.

4. Page 9, line 8: Reference #47 does not seem to show DDI1/2's function in removal of non-covalent replication barriers as stated here. The referred paper suggested that RTF2 is a replisome component that needs to be degraded in response to replication stress, but did not claim that RTF2 is a replication barrier.

Answer: The reviewer is correct. We have modified this.

5. Page 13: The section of "DPC proteases and DNA replication" is well-written, but the role of the proteasome (<https://doi.org/10.1016/j.molcel.2018.11.024>) is not mentioned.

Answer: We purposely excluded the proteasome from "DPC proteases and DNA replication" because we had dedicated a separate paragraph to the proteasome and cited the study brought up by the reviewer.

6. Page 14: line 22: Contrary to what is discussed here, Reference #58 seems to show CldU tract length to assess replication speed.

Answer: The reviewer is correct. We have reorganised this part.

7. Page 24, line 7: "Analogous phenotypes are observed after exposure of human cells to FA or CPT, indicating that they are due to defective DPC repair." This might not be a valid

Point-by-point response to reviewers

argument. FA causes DNA crosslink damage and CPT might generate dsDNA ends, both of which do not involve DPCs but could cause phenotypes analogous to SPRTN depletion.

Answer: We agree the conclusion might have been too strong. We have changed the sentence.

Minor points

1. Page 3, line 22: change “nucleases” to “proteases”

Answer: We have corrected this mistake.

2. Page 7, line 20 – Page 8, line 13: This part seems to be out of place. The authors might consider combining with the following section of “Emerging DPC proteases.”

Answer: We agree. This entire paragraph has been rearranged.

3. Page 8, line 19: Protease names are indicated in bold (here ACRC/GCNA), but it does not seem to be following a unified rule. ACRC/GCNA shows up again in bold in Page 11, and other proteases are indicated in bold in Page 12 for the second time.

Answer: We have restructured the organisation of all subtitles throughout the text to be uniform.

4. Page 23, line 2: Consider revising the sentence. (Rescue of) the replication defect depends on ...

Answer: We have taken the reviewer’s suggestion.

5. Page 26, line 3: replace “than” with “that”

Answer: We have corrected this mistake.

Reviewer #3 (Remarks to the Author):

In the manuscript, Ruggiano and Ramadan summarize latest advances in the field of DNA – protein crosslink (DPC) repair by proteases in relation to genome stability. The article begins with a general introduction on DPCs and follows by discussion on recently characterized DPC proteases and their known regulatory mechanisms. Thereafter, the review brings about recent important observations on the DPC-related proteases GCNA, SPRTN, and FAM11A in regulating DNA replication and genome stability across species, as well as the roles of SPRTN and FAM11A against DPC during DNA replication. Beyond the DNA replication, the authors also elaborate on reported roles of DPC proteases on mitosis, transcription, and other known non cross-linked substrates (i.e. histones, PARP1, and CHEK1).

Overall, the review is well structured and provides concise update on the current knowledge on DPCs and DPC proteases.

However, in addition to further proofreading, the present review article may benefit from the following:

Point-by-point response to reviewers

1. To include further description of DPC repair by NER on page 6, prior to the "Wss1 and SPRTN: the founding members of a class of repair enzyme"

Answer: We have not included an extensive description but have clarified that some DPC repair pathways work "by excising the DNA flanking the DPC". We have also added references:

Hoa *et al.*, Mre11 is essential for the removal of lethal Topoisomerase 2 covalent cleavage complexes, *Mol Cell* 2016.

Deshpande *et al.*, Nbs1 converts the human Mre11/Rad50 nuclease complex into an endo/exonuclease machine specific for protein-DNA adducts, *Mol Cell* 2016.

6. To highlight the emerging post-translational regulatory aspects (i.e. SUMO and ubiquitin) of DPC repair (i.e. page 19 - 21). Authors may also consider to incorporate the discussion together with discussion DPC proteases structure and activation by DNA on page 6 & 7

Answer: We have moved the entire discussion on PTMs in a separate paragraph.

2. To improve Figure 2 in illustrating different cellular aspects influenced by DPC is highly recommended

Answer: We have taken the reviewer's advice. Previous Figure 2 has been modified and moved to Figure 1 now.

3. To check and remove duplicated reference from the review - e.g. ref 74 & 98 are duplicate

Answer: We have corrected this mistake and double-checked all references.

REVIEWERS' COMMENTS:

Reviewer #1 (Remarks to the Author):

I have no further comments. I support publication.

Reviewer #2 (Remarks to the Author):

The authors addressed all of my comments and the manuscript is now suitable for publication in Communications Biology. I would like to congratulate the authors for the insightful review article.

Reviewer #3 (Remarks to the Author):

The authors have done a great job in incorporating the suggestion. This is a very interesting and clear review article.